# Mental and Sexual Health of Men in Times of COVID-19 Lockdown

**DOI:** 10.3390/ijerph192215327

**Published:** 2022-11-20

**Authors:** Ewa Szuster, Anna Pawlikowska-Gorzelańczyk, Paulina Kostrzewska, Amanda Mandera-Grygierzec, Agnieszka Rusiecka, Małgorzata Biernikiewicz, Kinga Brawańska, Małgorzata Sobieszczańska, Krystyna Rożek-Piechura, Dariusz Kałka

**Affiliations:** 1Cardiosexology Students Club, Wroclaw Medical University, 50-368 Wroclaw, Poland; 2Statistical Analysis Centre, Wroclaw Medical University, 50-367 Wroclaw, Poland; 3Studio Słowa, 50-357 Wroclaw, Poland; 4Clinical Department of Geriatrics, Wroclaw Medical University, 50-369 Wroclaw, Poland; 5Faculty of Physiotherapy, Wroclaw University of Health and Sport Sciences, 51-612 Wroclaw, Poland; 6Men’s Health Centre in Wrocław, 53-151 Wroclaw, Poland

**Keywords:** COVID-19 pandemic, mental health, sexual functioning, erectile dysfunction, men’s health, Polish population

## Abstract

Due to the worldwide spread of COVID-19, some restrictions were introduced which could lead to the development of distress and somatic symptoms. This survey aimed to study the mental and sexual health of men during the COVID-19 outbreak. An online questionnaire was conducted to collect data on contact with people suspected of infection/infected with the SARS-CoV-2 virus, use of stimulants, and perceived mental and sexual health during isolation among Polish men. They were also asked to answer the Beck Depression Inventory (BDI) and the International Index of Erectile Function (IIEF-15) questionnaire. In total, 606 men with a mean age of 28.46 ± 9.17 years took part in the survey. Fear of contracting the COVID-19 infection had a negative impact on the mental health of 132 men (21.8%). Fear of the health condition of loved ones caused stress and a depressed mood in 253 men (41.7%), and media reports worsened the mental health of 185 men (30.2%). In the BDI, 71.95% of the respondents did not suffer from depressive symptoms, 17.33% were diagnosed with mild depression, 6.11% with moderate depression, and 4.62% had severe depression. The mean score in the IIEF-15 questionnaire in the erectile function domain was 22.27, orgasm—7.63, desire—8.25, satisfaction—10.17, and general satisfaction—6.84. Depressive symptoms indicated more severe sexual functioning disorders (*p* < 0.001). Fear, following the media, and loneliness were associated with more severe depressive and sexual disorders (*p* < 0.001). The libido level (*p* = 0.002) and frequency of sexual activity (*p* < 0.001) were also lower during the pandemic than before the lockdown. These data showed that the COVID-19 pandemic had a significant impact on male mental and sexual health.

## 1. Introduction

The COVID-19 pandemic has caused changes in the daily functioning of people from all over the world. On 25 March 2020, the Polish government instated a lockdown. Travelling, meeting other people, and outdoor activities were forbidden. To protect citizens, shops were closed and remote work and home-schooling were suggested. These restrictions were introduced in a short period, without any premises and time to prepare or get used to the new situation. Moreover, the scale of the COVID-19 infection spread was terrifying. From 4 March 2020, 6,213,262 infections and 117,252 fatal SARS-Cov2 infection cases were reported in Poland [1]. Recent studies have shown that the COVID-19 pandemic has had an impact not only on physical conditions but also on mental health. Many researchers revealed a higher presence of depressive symptoms and higher levels of anxiety and stress [2,3,4]. Many young people faced stressful situations, such as fear of infection as well as job loss, suspension of education, and isolation. The COVID-19 pandemic had a significant impact on mental health, the economy, education, and relationships.

According to the World Health Organization, sexual health is a state of physical, emotional, mental, and social well-being regarding sexuality. It is influenced by a complex web of factors ranging from sexual behavior, attitudes, and societal factors, to biological risk and genetic predisposition [5]. Reduced physical activity, increased physiological stress, and reduced entertainment impacted sexual activity and functioning [6]. Due to the pandemic, many people were forced to be with their partners or to be isolated. This can emotionally influence relationships and sexual activity. [7] Furthermore, reports from the literature suggest that among young people, depression and anxiety are widespread. About 350 million people suffer from depression worldwide, while in Poland, this number reaches about 1.5 million [8]. Depression is also strongly related to erectile dysfunction [9]. Moreover, satisfaction with one’s sex life significantly reduces the level of stress and anxiety in many populations [10]. The study of Ibarra et al. showed that quarantine affects people’s emotions and decreases libido levels [11]. Lehmiller et al. conducted a study on changes in people’s sexual lives after the onset of the pandemic. The researchers reported that the frequency of sexual behaviors decreased compared to past year frequencies [12].

Having analyzed the phenomenon in which emotional distress could cause physical symptoms, we conducted a study to investigate the link between the COVID-19 outbreak and the mental and physical health of the male population.

## 2. Materials and Methods

We conducted an online survey on social media. The questionnaire collected data on COVID-19 infection or being in quarantine, contact with people suspected of COVID-19 infection/infected people, fear of COVID-19 infection, use of stimulants, and perceived mental and sexual health. Sexually active adults aged 18 years or older, currently residing in Poland and self-isolating/social-distancing due to COVID-19, were eligible to participate. All men were asked to complete the Beck Depression Inventory (BDI) and the International Index of Erectile Function (IIEF-15) questionnaire. The questionnaire included additional demographic questions. The study was conducted from 22 April 2020 through 7 May 2020. Data collection started one month after the lockdown in Poland was established (25 March 2020). 

Mental health was investigated using the Beck Depression Inventory (BDI) questionnaire, a 21-question multiple-choice self-report inventory, which measures attitudes and characteristic symptoms of depression [13]. Each question is scored on a scale of 0–3. Total scores can range from 0 to 63. Standard cut-off scores are 0–11 minimal depression, 12–19 mild depression, 20–25 moderate depression, and 26–63 severe depression.

The International Index of Erectile Function (IIEF-15) questionnaire is a patient-reported outcome measure widely used to measure erectile dysfunction (ED). Each answer is scored on a discrete scale value of 0–5 (questions 1–10) and 1–5 (questions 11–15). The 15 questions examine the 5 main domains of male sexual function over the last four weeks: erectile function, orgasmic function, sexual desire, intercourse satisfaction, and overall satisfaction [14]. Erectile function scores of the IIEF-15 total up to 30 points. ED was defined as a score of 25 or less. All collected questionnaires were analyzed. Our participants were informed that participation was voluntary and that they would remain anonymous. Every man provided informed consent to participate in the study. The questionnaire was verified with the CHERRIES checklist [15]. The study was approved by the Commission of Bioethics at Wroclaw Medical University, Wroclaw, Poland (KB-424/2021).

Data were statistically analyzed using Statistica software v. 13.3 (StatSoft, Tulsa, OK, USA). The data were presented as numbers, percentages, and means with standard deviations. The Shapiro–Wilk test was used to analyze the distribution of the data. The Chi-square test was used to compare sexual activity frequency and libido before and during the pandemic. To compare all the variables between groups, the Mann-Whitney U test was used. The Kruskal–Wallis test with the post hoc median test was used when comparing more than two continuous variables. Spearman’s rank correlation coefficient was used to measure the strength of the association between the FSFI and BDI scores. The differences were interpreted as statistically significant at *p* < 0.05. Cronbach’s alpha was used to assess the internal consistency of the questionnaire. A value higher than 0.7 indicates good internal consistency. This indicator was calculated for the entire questionnaire, covering all questions from the BDI (21 questions) to the IIEF (15 questions) sections. The results indicate that the questionnaire had good overall internal consistency, with a Cronbach’s alpha of 0.83. Moreover, internal consistency was assessed separately for both parts of the questionnaire. The Cronbach’s alpha for the BDI and IIEF parts was 0.92 and 0.93, respectively.

## 3. Results

In our analysis, we included 606 men with a mean age of 28.46 ± 9.17 years old. Our respondents were mostly employed (working in the workplace), in a partnership, and living in big cities. The characteristics of the study group are presented in Table 1.

Of our respondents, 11.39% had comorbid chronic diseases, and 17.82% were on treatment due to a disease. In total, 2.31% were in quarantine, while 14.36% had friends or family in quarantine. Only one participant was diagnosed with COVID-19. Nevertheless, 5.61% had friends/family infected with the SARS-Cov2 virus, and 7 participants (1.16%) stated they had experienced the death of a friend/family member due to COVID-19.

In our study, 20 men (3.30%) were under psychiatric/psychological care during the COVID-19 pandemic, while only half (1.65%) used sedatives. In the BDI, 71.95% of respondents did not reveal depressive symptoms; 17.33% were diagnosed with mild depression; 6.11% were diagnosed with moderate depression, and 4.62% had severe depression. The detailed questions concerning the psychological condition of our respondents are presented in Table 2.

The libido level was lower during the COVID-19 pandemic than before the pandemic (*p* = 0.002). Consequently, the frequency of sexual activity was also significantly lower during the pandemic than before the lockdown (*p* < 0.001).

The mean score in the IIEF-15 questionnaire in the erectile function domain was 22.27, which should be interpreted as mild erectile dysfunction. The mean scores in orgasm, desire, and satisfaction were also related to mild dysfunction. General satisfaction corresponded to moderate-mild dysfunction. The detailed IIEF-15 results are presented in Table 3.

Our findings also revealed that depressive symptoms indicated more severe sexual functioning disorders (*p* < 0.001). What is more, fear, following the media, and loneliness also had a negative impact on depressive and sexual disorders (*p* < 0.001). Nevertheless, older men achieved better IIEF-15 scores but had more severe depressive symptoms (*p* < 0.001). The use of stimulants (alcohol/cigarettes) was higher among men with more severe sexual functioning disturbances. The detailed data are presented in Table 4 and Table 5.

We found no correlation between the education level (*p* = 0.31) or the city size (*p* = 0.52) with sexual functioning. In the ANOVA Kruskal-Wallis test, we found that respondents in relationships (married or in informal relationships) had significantly better sexual function than single men (*p* < 0.001). Furthermore, we have also found no correlation between education (*p* = 0.25) or city size (*p* = 0.91) and depression. Using the ANOVA Kruskal-Wallis test, we found that respondents in relationships had a significantly lower risk of depressive disorders than single men (*p* < 0.001). In addition, married men had a significantly lower risk of depression than men in informal relationships (*p* < 0.001). 

We have also found that men on pharmacotherapy scored lower in IIEF-15 erection (*p* = 0.01) and desire (*p* = 0.02) domains. Furthermore, the respondents under psychiatric/psychological supervision, due to the pandemic, achieved lower scores in the general satisfaction domain (*p* = 0.04).

## 4. Discussion

Our study aimed to investigate the link between the COVID-19 outbreak and the mental and physical health of the male population. We found that of 606 surveyed men, 21.8% confirmed that fear of contracting the COVID-19 infection had a negative impact on their mental health, 41.7% confirmed that fear of the health condition of loved ones caused stress and a depressed mood, and 30.2% confirmed that media reports worsened their mental health. Overall, 28.05% of surveyed men experienced depressive symptoms. The erectile function domain score from the IIEF-15 questionnaire was 22.27 showing mild erectile dysfunction in the study group. The occurrence of symptoms of depression was significantly correlated with the occurrence of severe sexual functioning disorders. The intensity of other fears was also correlated with depressive and sexual disorders.

Stress related to the COVID-19 pandemic and fear of infection promote a weakening of the immune system [16]. There is evidence supporting the thesis that sexuality supports the immune system; therefore, maintaining an acceptable frequency of sexual intercourse is highly recommended. Healthy sexual activity improves the immune system and also helps soothe the negative psychological effects associated with COVID-19 infection [17]. The results of this study revealed the impact of the COVID-19 pandemic on human health. Many scientists note a correlation between lockdowns, fear of coronavirus infection, and higher prevalence of depressive symptoms. In the literature review conducted by Burmistova et al., the findings showed that almost 18% of the population experienced at least mildly severe anxiety symptoms, 18.85% exhibited at least mild symptoms of depression, and 36.59% complained about sleep disturbances [18]. 

Similar results were seen in our study, as 17.33% of respondents suffered from mild depression. The COVID-19 pandemic significantly reduced interpersonal contact by maintaining social distance. Accordingly, sexual contact also appeared to be involved in the transmission of the virus, as the infection is spread by aerosolized particles [19]. According to available knowledge, the virus can persist on surfaces and clothes for several days, and infected individuals can spread respiratory secretions to their partner’s skin and personal belongings. However, data on other routes of transmission are lacking [19]. It can be concluded that the COVID-19 pandemic had a significant impact on sexual health. Fang et al. reported that the COVID-19 pandemic caused a deterioration in sexual function associated with increased depression, anxiety, and a decreased sex life frequency [20]. 

Louis et al. reported that in Great Britain, during the first lockdown, 39.9% of women and men engaged in sexual activity at least once a week [21]. According to the results of our study, more than half of the respondents (55.2%) engaged in sexual activity at least once a week. Cocci et al. showed that more than 40% of the study group admitted an increased sexual desire during the quarantine, compared to the pre-pandemic situation. However, enhanced desire did not contribute to the increase in the frequency of sexual intercourse [22], whereas our study showed that the desire among the respondents corresponded to mild dysfunctions.

Many researchers investigated sexual desire before the pandemic and during the lockdown. Ballester-Arnal et al.’s study conducted in Spain shows that 39,5% of participants declared greater sexual desire during the lockdown. On the other hand, 34,9% declared lower sexual desire [23]. Pancerei et al.’s study in Italy revealed that 18,20% of men felt decreased sexual desire [24]. Another study conducted in the USA shows that 25% of respondents reported less sexual desire, and 18% of men increased sexual desire [25]. Masoudi et al. perform a systematic literature review. Most of the mentioned studies show a decreased number of sexual relationships during and prior to the COVID-19 pandemic, and the reduction in sexual intercourse was significant. Moreover, other studies revealed other factors influencing participants’ sexual activity and functioning, such as alcohol use, high number of self-isolation days, having children at home, and lack of privacy. Also, three studies investigated male sexual functioning using the IIEF-5 questionnaire and revealed a statistically significant difference before and during the pandemic [26]. In addition, the finding of a study conducted by Karagöz et al. using the IIEF-15 questionnaire showed statistically significantly lower results of male sexual function during the COVID-19 pandemic compared to the pre-pandemic period. However, it was also concluded that there was no significant decrease in male sexual desire [27]. Based on the findings of an Italian study during the COVID-19 pandemic among healthcare workers, De Rose et al. showed that low sexual satisfaction was a predictor of a low level of sexual desire [28].

In addition, the reasons for ED are varied. ED can be caused by hypertension, diabetes mellitus, obesity, hyperlipidemia, and testosterone deficiency. It is also important to emphasize the physiological causes such as depression, anxiety, and problems in the relationship. Furthermore, many drugs and substances can contribute to the development of ED [29]. During the COVID-19 pandemic, many symptoms were exacerbated. The feeling of loneliness, fear of coronavirus infection and death, and more severe symptoms of depression had an impact on men’s sexual health. Karkin and Alma also revealed changes in endothelium, in corpus cavernosum biopsy, in patients who had developed COVID-19 [30]. Also, the testosterone level was significantly lower after COVID-19 infection [31]. 

The risk factors of COVID-19 infection and ED are strongly associated. Katz et al. showed a correlation between the incidence of COVID-19 and the occurrence of ED. Moreover, smokers were reported to be disproportionately affected by both conditions [32].

Disturbances in mental and sexual health could have negative effects on daily functioning. Physicians, sexologists, psychologists, and governments should pay attention to individuals who present depressive symptoms or sexual dysfunction. General practitioners are also very important. They should be able to recognize symptoms early and implement cures. What is more, social campaigns should educate people about methods of dealing with social isolation, feelings of loneliness, and feelings of fear. Furthermore, healthcare professionals should inform patients that mental and sexual dysfunction is common in order to decrease the feeling that one is isolated with their problems. Additionally, people should be enhanced to check the information from reliable, official sources. Physical activity also affects mental health, and sports and outdoor activities should be suggested and popularized [33].

Several limitations of this study should be kept in mind while interpreting the results. Due to the lockdown, the questionnaire had to be published online. As a result, the study is based on cross-sectional data. Consequently, our findings are not representative of the entire male population. More research is required on a much larger scale to understand the impact of the COVID-19 pandemic on physical and reproductive health in different populations and regions.

## 5. Conclusions

The COVID-19 pandemic has had a significant impact on the mental and sexual health of many people. The results of many studies show a link between mild depressive disorder and ED caused by the lockdown. Therefore, it is extremely important to undertake interventions aimed at improving health and overall well-being. In addition, healthcare and society have focused on combating the COVID-19 pandemic, and therefore basic reproductive health services have been disrupted.

## Figures and Tables

**Table 1 ijerph-19-15327-t001:** Sociodemographic characteristics of the group.

Variable	N = 606
Age, years (distribution other than normal)	Mean 28.46 ± 9.17
	Median 25.00
	IQR (22–31)
	Range (18–76)
Education	
Primary	13 (2.15%)
Vocational	36 (5.94%)
Secondary	248 (40.92%)
Higher	309 (50.99%)
Employment status	
Employed—working at the workplace working students	245 (40.43%) 17 (2.81%)
Remote work working students	180 (29.70%) 29 (4.79%)
Employment issues due to the COVID-19 pandemic	13 (2.15%)
Sick leave	14 (2.31%)
Unemployed due to other reasons	9 (1.49%)
Full-time student	139 (22.94%)
Student-income lost	2 (0.33%)
Pensioner	4 (0.66%)
Marital status	
Single	146 (24.09%)
Married	127 (20.96%)
In partnership	333 (54.95%)
Place of living	
Rural area	103 (17.00%)
City > 50,000 inhabitants	83 (13.70%)
City from 50,000 to 100,000 inhabitants	40 (6.60%)
City from 100,000 to 250,000 inhabitants	81 (13.37%)
City above 250,000 inhabitants	299 (49.34%)

**Table 2 ijerph-19-15327-t002:** Attitudes and habits related to the pandemic.

Variable	N = 606
Fear of infection with coronavirus has a negative impact on my mental health	
Strongly agree	26 (4.29%)
Agree	106 (17.49%)
Undecided	101 (16.67%)
Disagree	162 (26.73%)
Strongly disagree	211 (34.82%)
Fear of the heath condition of the loved ones is a source of stress and depressed mood	
Strongly agree	50 (8.25%)
Agree	203 (33.50%)
Undecided	104 (17.16%)
Disagree	130 (21.45%)
Strongly disagree	119 (19.64%)
Following the media reports is a source of a significant deterioration of my mental state	
Strongly agree	67 (11.06%)
Agree	118 (19.47%)
Undecided	105 (17.33%)
Disagree	137 (22.61%)
Strongly disagree	179 (29.54%)
Perceived loneliness caused by isolation from the world / loved ones	
Strongly agree	108 (17.82%)
Agree	185 (30.53%)
Undecided	84 (13.86%)
Disagree	121 (19.97%)
Strongly disagree	108 (17.82%)
More frequent use of alcohol/cigarettes cause by pandemic	
Strongly agree	56 (9.24%)
Agree	97 (16.01%)
Undecided	52 (8.58%)
Disagree	110 (18.15%)
Strongly disagree	291 (48.02%)

**Table 3 ijerph-19-15327-t003:** Sexual activity and libido levels before and during the pandemic.

Frequency of Sexual Activity before/during the Pandemic	Before	During
	*p* < 0.001	
Several times a day	27 (4.46%)	38 (6.27%)
Every day	72 (11.88%)	67 (11.06%)
Several times a week	281 (46.37%)	224 (36.96%)
Once a week	71 (11.72%)	62 (10.23%)
Several times a month	90 (14.85%)	77 (12.71%)
Once a month	17 (2.81%)	27 (4.46%)
Fewer than once a month	48 (7.92%)	111 (18.32%)
Libido level before/during the pandemic	before	during
	*p* = 0.002	
High	308 (50.83%)	266 (43.89%)
Moderate	264 (43.56%)	276 (45.54%)
Decreased libido	34 (5.61%)	64 (10.56%)

**Table 4 ijerph-19-15327-t004:** International Index of Erectile Function 15 results.

Domain	Score, Mean ± SD	Range
Erection	22.27 ± 10.21	0–30
Orgasm	7.63 ± 3.91	0–10
Desire	8.25 ± 1.79	0–10
Satisfaction	10.17 ± 4.28	0–15
General satisfaction	6.84 ± 3.08	0–10

**Table 5 ijerph-19-15327-t005:** Correlations between Beck Depression Inventory scores, International Index of Erectile Function-15 scores, and other variables (Spearman’s rank correlation coefficient).

Variable	Variable	Correlation Coef.	*p* Value
BDI	IIEF-15 score	−0.469	<0.001
IIEF	Age	0.225	<0.001
	Fear of infection	−0.161	<0.001
	Fear of health condition	−0.137	<0.001
	Following the media	−0.181	<0.001
	Perceived loneliness	−0.291	<0.001
	More frequent use of alcohol/cigarettes	−0.215	<0.001
BDI	Age	0.251	<0.001
	Fear of infection	0.303	<0.001
	Fear of health condition	0.225	<0.001
	Following the media	0.258	<0.001
	Perceived loneliness	0.321	<0.001
	More frequent use of alcohol/cigarettes	−0.079	0.05

BDI, Beck Depression Inventory; IIEF-15, International Index of Erectile Function-15. Strongly agree = 5; strongly disagree = 1.

## Data Availability

Data are contained within the article.

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
