# Peer review of "Mental and Sexual Health of Men in Times of COVID-19 Lockdown"

_ijerph, 2022, doi:10.3390/ijerph192215327_

Round 1
Reviewer 1 Report
The article examines a fascinating phenomenon: effects of the COVID on sexual function.
Please propose (you may do that in the summary) to examine whether it is possible to define the phenomena with what the literature called "unrecognized loss" (loss of sexual normative function) - a phenomenon that should be perceived as part of those phenomena that it is desirable for social science researchers to examine in the perspective of the "Politics of Recognitions". After all, until this is not recognized - the phenomenon will remain in the private sphere, will lead people to feel shame, delegitimize to express themselves in relation to it, and will not be recognized as an indispensable part of the side effects of the COVID. Which will also prevent research, prevention, and treatment.
For the benefit of being mentioned as a reference to the phenomenon, which indicates the connection between the mental state and the lack of recognition, the bibliographic item should be mentioned:
Lebel, U., " 'Second Class Loss': Political Culture as a Recovery Barrier? – Israeli Families of Terrorist Casualties and their Struggle for National Honors, Recognition and Belonging", Death Studies, 38(1), 2014, 9-19.
Reviewer 2 Report
The manuscript entitled "Mental and Sexual Health of Men in Times of COVID-19 Lockdown" is an interesting study but has a lot of deficiencies. Below is a list of my suggestions for improving the manuscript:
-The introduction is too general and does not introduce the reader to the problems and goals of the research. The paper lacks an adequate literature review. Given the significance and timeliness of the problem, the references used are insufficient and do not demonstrate fundamental scientific discoveries in the field of public health from the observed perspective.
- The methodology is not in accordance with the general principles of writing a scientific paper. The authors didn't say enough about the measurement scales they used, and they didn't use the right statistical methods to test how reliable they were.
- The results are very descriptive and are not presented at the level of a scientific paper worthy of publication in an international scientific journal.
- Furthermore, the shortcomings of the work are its discussion of results, conclusions, and practical contributions. The authors should illustrate how the results compare to similar research done in the region or in other countries as part of the discussion. What are the theoretical and practical implications of the paper? In conclusion, the authors should present the implications for public health as well as for men's mental health caused by the COVID-19 Lockdown.
Reviewer 3 Report
The paper is interesting and novel, and it accounts for the importance of sexual and mental health in men in Poland.
However, I would like to highlight some areas of opportunity that should be considered by the authors.
1. It is mentioned that a questionnaire on sociodemographic data, concerns, fear, habits, among others, was designed in the first phase of the COVID-19 pandemic, but the authors do not describe this questionnaire or the way in which it was validated.
2. Regarding the selection of the participants, they do not refer to the type of sampling or selection of the participants, they only refer that it was disseminated through social networks.
3. They refer that the application of the survey was through digital channels. Kindly refer to the checklist for Reporting Results of Internet E-Surveys (CHERRIES) and include it in your method section.
4. Considering that it was applied in the first phase of the pandemic declaration between April and May, little was known about the real risks of infection and they describe that only a small proportion of the participants were not in social isolation (Table 1).
5. Perhaps the authors could explain how the follow-up of the recommendations to contain the pandemic was experienced in the country and be able to associate it with the variables studied.
6. In the results, the authors report that they found results indicating that almost a third of the sample was affected by moderate to severe levels of depression and little affectation of sexual dysfunction among them. However, they report that these variables were associated to a greater extent among men who had depression and sexual impairment at the beginning of the pandemic compared to another stage of the pandemic, but it is not clear whether they compared it or not.
7. You report that the level of libido was lower at the beginning of the pandemic than at the end of the pandemic, how did you calculate this difference? It is not presented if they evaluated the variables at another time.
8. Table 2 has as title sociodemographic characteristics of the sample and what is presented in the content are other variables.
9. It would be interesting to present a comparative analysis between participants with and without mental health problems and the other study variables, in addition to presenting the effect size of the difference between the measures compared.
10. It is also mentioned that associations between variables were found but no regression analysis was calculated, how did you reach this conclusion?
11. Revise the discussion based on the corrections of the statistical analyses.
12. Mention the limitations of the study
Round 2
Reviewer 2 Report
The authors mostly addressed the reviewers' recommendations. The paper, in my opinion, is suitable for publishing in the IJERPH.